# National Information Campaign Revealed Disease Characteristic and Burden in Adult Patients Suffering from Atopic Dermatitis

**DOI:** 10.3390/jcm11175204

**Published:** 2022-09-02

**Authors:** Niccolò Gori, Andrea Chiricozzi, Franco Marsili, Silvia Mariel Ferrucci, Paolo Amerio, Vincenzo Battarra, Salvatore Campitiello, Antonio Castelli, Maurizio Congedo, Monica Corazza, Antonio Cristaudo, Gabriella Fabbrocini, Giampiero Girolomoni, Giovanna Malara, Giuseppe Micali, Giovanni Palazzo, Aurora Parodi, Annalisa Patrizi, Giovanni Pellacani, Paolo Pigatto, Eugenio Provenzano, Pietro Quaglino, Marco Romanelli, Mariateresa Rossi, Paola Savoia, Ketty Peris

**Affiliations:** 1UOC di Dermatologia, Dipartimento di Scienze Mediche e Chirurgiche, Fondazione Policlinico Universitario Agostino Gemelli-IRCCS, 00168 Rome, Italy; 2Dermatologia, Dipartimento di Medicina e Chirurgia Traslazionale, Università Cattolica del Sacro Cuore, 00168 Rome, Italy; 3Dermatology Unit, Versilia Hospital, ASL 12, 55049 Lido di Camaiore, Italy; 4Dermatology Unit, Fondazione IRCCS Ca’ Granda Ospedale Maggiore Policlinico, 20122 Milan, Italy; 5Dermatologic Clinic, Department of Medicine and Aging Science, University G.d’Annunzio, 66100 Chieti, Italy; 6Unit of Dermatology, AORN Sant’Anna e San Sebastiano, 81100 Caserta, Italy; 7U.O.C. Dermatologia, ASL Salerno, Ospedale “A:Tortora”-Pagani, 84016 Salerno, Italy; 8Dermatology Unit, San Donato Hospital, ASL 8, 52100 Arezzo, Italy; 9Section of Dermatology, Vito Fazzi Hospital, Piazza Filippo Muratore, 73100 Lecce, Italy; 10Section of Dermatology and Infectious Diseases, Department of Medical Sciences, University of Ferrara, 44121 Ferrara, Italy; 11Istituto Dermatologico San Gallicano IRCSS, IFO, 00100 Rome, Italy; 12Department of Clinical Medicine and Surgery, University of Naples Federico II, 80131 Naples, Italy; 13Section of Dermatology and Venereology, Department of Medicine, University of Verona, 37129 Verona, Italy; 14Struttura Complessa di Dermatologia, Grande Ospedale Metropolitano ‘Bianchi Melacrino Morelli’, 89129 Reggio Calabria, Italy; 15Dermatology Clinic, University of Catania, 95123 Catania, Italy; 16Ambulatorio di Dermatologia, Ospedale Distrettuale di Tinchi, 75015 Pisticci, Italy; 17DiSSal Section of Dermatology, University of Genoa-Ospedale-Policlinico San Martino IRCCS, 16132 Genoa, Italy; 18Division of Dermatology, Department of Experimental, Diagnostic and Specialty Medicine, University of Bologna, 40138 Bologna, Italy; 19Dermatology Clinic, Department of Clinical Internal, Anesthesiological and Cardiovascular Sciences, Sapienza University of Rome, 00161 Rome, Italy; 20Department of Medical, Surgical and Odontoiatric Science, IRCCS Ospedale Ortopedico Galeazzi, 20161 Milan, Italy; 21Unit of Dermatology, Mariano Santo Hospital, 87100 Cosenza, Italy; 22Dermatologic Clinic, Department of Medical Sciences, University of Turin, 10121 Torino, Italy; 23Dermatology Department, University of Pisa, Via Roma 67, 56126 Pisa, Italy; 24Department of Dermatology, University of Brescia, 25121 Brescia, Italy; 25Department of Health Science, University of Eastern Piedmont, Via Solaroli 17, 28100 Novara, Italy

**Keywords:** atopic dermatitis, information campaign, early diagnosis

## Abstract

Atopic dermatitis (AD) is a common inflammatory skin disease often associated with a significant impairment in the quality of life of affected patients. The Italian Society of Dermatology and Venereology (SIDeMaST) planned a national information campaign, providing direct access to 27 dermatologic centers dedicated to the management of AD. The aim of this study aimed was to outline critical aspects related to AD in the general population. Overall, 643 adult subjects were included in this study, and in 44.2% (284/643) of cases, a diagnosis of AD was confirmed, whereas about 55% of subjects were affected by other pruritic cutaneous diseases. Higher intensity of pruritus and sleep disturbance, as well as an increased interference in sport, work, and social confidence was reported in the AD group compared to the non-AD group. In the AD subgroup, the mean duration of disease was of 15.3 years, with a mean eczema area and severity index (EASI) score of 11.2, and investigator global assessment (IGA) score of 1.9 and an itch numeric rating scale (NRS) of 6.9. Almost 32% of patients were untreated, either with topical or systemic agents, whereas 44.3% used routine topical compounds (topical corticosteroids and calcineurin inhibitors), and only 7.0% of patients were systemically treated. Only 2.8% of patients reported complete satisfaction with the treatment received for AD to date. This study reveals a profound unmet need in AD, showing a poorly managed and undertreated patient population despite a high reported burden of disease. This suggests the usefulness of information campaigns with the goal of improving patient awareness regarding AD and facilitating early diagnosis and access to dedicated healthcare institutions.

## 1. Introduction

Atopic dermatitis (AD) is the most common chronic inflammatory skin disease, affecting nearly 230 million people worldwide, with a prevalence, in developed countries, ranging between 10% and 25% in children and 7–10% in adults [1]. AD is clinically characterized by intense itch, dry skin, and eczematous lesions, with the preferential involvement of flexures, head, neck, and hands in adulthood [1]. It is frequently associated with a personal and/or family history of atopic extracutaneous manifestations, such as allergic rhinitis, conjunctivitis, and asthma [1]. Several non-atopic diseases, including inflammatory, autoimmune, and mental health disorders, might be also observed in AD patients [2]. In addition, attention disturbances and poor sleep quality, likely related to itch, may occur, affecting school and work performance [3]. Treatment of mild AD is essentially based on the use of topical corticosteroids (TCSs) and calcineurin inhibitors (TCIs), whereas moderate-to-severe AD, accounting for as many as one-third of cases, is commonly treated with phototherapy and/or systemic therapies, including traditional immunosuppressants and novel immune-targeted therapies [4,5]. In addition, to improve skin dryness and reduce itch, moisturizers are usually applied daily, implying a significant economic burden for patients [6]. Notably, a recent cross-sectional study including nine European countries reported a mean annual personal extra out-of-pocket expense of EUR 927.12 for patients with AD [6].

Although they were conceived in the 1980s, the Hanifin and Rajka criteria are still the most used tool to diagnose AD in both clinical practice and research settings, whereas, considering the lack of specific diagnostic markers, diagnosis of AD is essentially based on the accurate evaluation of clinical signs, symptoms, and medical history by skilled physicians [7]. Diagnosis of AD is relatively easy in children but often challenging in adults, especially in late-onset forms, due to a broader clinical variability [7,8]. For this reason, adult AD is thought to be underdiagnosed, reflecting the highly variable prevalence of disease reported in the literature, with an estimated range varying from 0.3% to 14.3% [8,9,10]. To reduce the proportion of undiagnosed and undertreated cases of AD, it might be helpful to promote awareness of AD through multiple strategies, including the diffusion of disease information websites and the organization of screening campaigns. Despite the existence of numerous scientific societies and patient associations providing educational websites for AD patients, the first national information and screening campaign was organized in September 2020 by the Italian Society of Dermatology and Venereology (SIDeMaST), which provided direct access to several dermatologic centers dedicated to the management of AD, with the aim of improving patient awareness about AD and facilitating early diagnosis and access to optimal treatment management.

## 2. Materials and Methods

In this study, we considered data obtained from subjects referred to 27 dermatology centers homogeneously distributed in northern, central, and southern Italy for the first national AD screening program supported by the Italian Society of Dermatology (SIDeMaST). The purpose of the program was to promote the knowledge of AD among adult subjects with established disease or suspected symptoms.

In September 2020, information regarding the national AD screening program was posted on the patient-oriented AD website (www.dallapartedellatuapelle.it). In particular, the web site provided general information regarding pathogenesis and clinical presentation of AD in adults, as well as email and telephone contacts to join the AD screening program. Patients with suspected or diagnosed AD were screened by non-physician personnel on the phone or by e-mail through a brief questionnaire. As a screening program, it was not necessary to apply for ethics committee approval because patients did not furnish any sensitive data to the centers.

Inclusion criteria for prescreened patients were limited to comprehension of written Italian language and consent to compile a printed survey. In each dermatological center, patients were required to complete a 21-item questionnaire about demographic and clinical data, including age, sex, weight, height, job, disease duration, type of medical specialists previously consulted, personal and family history of AD and comorbidities, interference of AD with physical activities and work tasks, therapeutic management of disease, economic burden for supplying topical and systemic drugs, and consultations for AD. Individuals were subdivided according to working profession as white collar (intellectual jobs, including doctors, lawyers, teachers, office workers, managers, and civil servants) or blue collar (manual jobs, including craftsmen, farmers, specialized workmen, drivers of industrial machines/vehicles, armed services, and unqualified professions) [11].

Patients were evaluated by dermatologists with experience in inflammatory skin diseases to assess the diagnosis of AD and suggest the most appropriate therapeutic approach. Disease severity in patients with a confirmed diagnosis of AD was assessed using (a) the eczema area severity index (EASI), with scored ranging from 0 to 72; (b) the investigator global assessment (IGA), with scores ranging from 0 to 4; (c) the itch numeric rating scale (NRS), ranging from 0 to 10, assessing itch intensity (itch-NRS); (d) a 0–10 NRS scale rating sleeplessness (sleep-NRS); (e) a 0–10 NRS evaluating disease-induced embarrassment (e-NRS); (f) a 0–10 NRS evaluating the influence of disease on work tasks (w-NRS); and (g) a 0–10 NRS assessing the impact of disease on sporting activity (s-NRS).

### Statistical Analyses

Categorical variables were analyzed as frequencies and percentages. Continuous variables were analyzed as mean and standard deviation (SD) or medians and interquartile ranges (IQRs). Variable normality was assessed by the Shapiro–Wilk W test. We compared questionnaire-obtained personal and clinical data between AD and non-AD groups, using a T test for comparison of means or Mann–Whitney test for comparison of medians, and by chi-square (or Fisher’s exact test) for categorical variables. In the subgroup with confirmed AD diagnosis, clinical data were described in terms of frequencies and percentages, whereas EASI and IGA scores were analyzed as mean and SD. All statistical tests were two-tailed, and a *p*-value less than 0.05 was considered statistically significant. Analysis was performed using STATA 17 software (StataCorp, College Station, TX, USA).

## 3. Results

### 3.1. Characteristic of the General Population

Overall, 641 adult subjects were referred to the 27 outpatient dermatology centers during the open day in September 2020. Demographic and clinical data are summarized in Table 1.

Work activity was classified as intellectual jobs (white collars), accounting for 40.6% of participants (260/641); or manual jobs (blue collars), accounting for 24.0% of participants (154/641); whereas 18.4% of participants (118/641) were students and 21.5% (138/641) were retired professionals. Proportions of 84.9% (544/641) and 34.5% (221/641) of participants reported to have visited a dermatologist an allergist, respectively, at least one time for their skin problems, whereas only 6.4% (41/641) were managed by their general practitioner.

The mean duration of skin manifestations was 10.7 ± 11.8 years. Mean itch-NRS and sleep-NRS values at the time of evaluation were 6.5 ± 2.5 and 4.4 ± 3.4, respectively. Approximately 55% (352/641) of patients reported itch as the major cause of discomfort, whereas another 17.3% (111/641) of subjects identified the presence of eczematous lesions as the main burdening factor, with 26.5% (170/641) reporting being equally disturbed by both manifestations.

### 3.2. Clinical Features of the AD Subpopulation Differ from Those of the Non-AD Population

A diagnosis of AD was confirmed in 44.3% (284/641) of the screened population.

On the other hand, 55.7% of subjects were included in the non-AD group, as they reported being affected by other pruritic cutaneous disease, including seborrheic dermatitis, psoriasis, allergic contact dermatitis, scabies, prurigo nodularis, dermatitis herpetiformis, bullous pemphigoid, and pruritus.

Demographic and clinical data of each subpopulation are summarized in Table 1. 

Approximately 77% (219/284) of AD patients had previously received a diagnosis of AD. In 35.3% (126/357) of non-AD patients, an erroneous diagnosis of AD had been previously made. Most AD patients (93.0%; 264/284) visited a dermatologist at least once, compared to 77.9% (278/357) of the non-AD population, whereas 48.9% (139/284) of AD compared to 20.4% (80/357) of non-AD patients had visited their skin disease evaluated by an allergologist and 2.1% (6/284) of AD versus 9.2% (33/357) of non-AD patients by a general practitioner. Mean duration of skin disease was 15.3 ± 12.6 years in the AD group and 7.0 ± 9.7 in the non-AD group (*p* < 0.0001). A family history of AD, rhino-conjunctivitis, and asthma was significantly more frequent in AD patients compared to non-AD patients (*p* < 0.0001). A higher intensity of pruritus and sleep deterioration were detected in the AD group, with mean values of itch- and sleep-NRS of 6.9 ± 2.4 and 5.0 ± 3.4, respectively, compared to 6.2 ± 2.6 and 3.9 ± 3.4 in the non-AD cohort (Table 1). Moreover, a higher grade of disease-related embarrassment and disease interference with sport and work was reported in the AD group compared with the non-AD population (Table 1). The regular use of TCS or TCI was reported in 44.3% (126/284) of AD patients compared to 37.2% (133/357) of non-AD patients (*p* < 0.0001). In addition, AD patients reported a greater use of moisturizers compared to non-AD patients (56.3% (160/284) versus 37.5% (134/357), *p* < 0.0001). On the contrary, no significant differences between the two populations were detected in terms of the use of systemic therapies. Only 2.8% (8/284) of AD patients and 4.5% (16/357) of the non-AD group reported complete satisfaction with therapy received, whereas 34.5% (98/284) of AD subjects and 18.8% (67/357) of non-AD sub-cohort reported only partial satisfaction, and 57.7% (164/284) of AD patients and 60.2% (215/357) of non-AD patients reported no satisfaction (*p* < 0.0001).

No significant difference was detected in terms of the average monthly expense for topicals, systemic drugs, and visits between the AD and non-AD groups (Table 2).

### 3.3. Physician-Oriented Assessment of AD Patients

In individuals with a confirmed diagnosis of AD (284 patients), mean EASI and mean IGA scores were 11.2 ± 12.0 and 1.9 ± 2.0, respectively. Moderate-to-severe AD, defined by an EASI score ≥16 and an IGA score ≥3, was reported in 21.5% (61/284) and 25.3% (72/284), respectively. The upper limbs were the body site more frequently affected by skin lesions (72.2%), followed by head and neck (49.6%), lower limbs (44,0%), trunk (28.2%), and back (25.7%). History of atopic comorbidities was reported in 48.2% of patients; in particular, rhinitis was described in 36.3% (103/284) of patients, conjunctivitis in 20.4% (58/284), and allergic asthma in 21.8% (62/284). In this patient population, the increase in disease severity scores was directly associated with a monthly expense for topical treatments of more than EUR 20, whereas no significant correlation between disease severity and monthly expense was observed for systemic drugs and visits (Table 2).

## 4. Discussion

AD is a chronic inflammatory skin disease associated with a significant deterioration of patients’ quality of life [1,3]. Although it is the most common inflammatory skin disease, the current lack of specific diagnostic markers and criteria makes the identification of adult AD challenging, particularly in the adult-onset subtype [7,12].

The latest national and international guidelines suggest that diagnosis of AD in adulthood is essentially clinical, based on evaluation of morphology and distribution of lesions and the exclusion of possible differential diagnoses, including allergic contact dermatitis, scabies, dermatitis herpetiformis, and cutaneous lymphomas [7,13]. The lack of experienced general practitioners and territorial dermatologists in recognizing adult AD could result in an underestimation of disease prevalence and burden. 

In this study, 44.3% (284/641) of the whole population received a clinical diagnosis of AD, and in 22.9% (65/284) of these cases, a different diagnosis was proposed during previous visits, most of which had been performed by dermatologists. 

Notably, more that 35% (126/357) of patients who resulted not affected by AD in this study had received an incorrect diagnosis of AD during previous visits, thus revealing not only a low sensitivity but also a low specificity in the diagnosis of adult AD with possible overestimation of disease prevalence in some cases. This significant number of misdiagnosedcases of AD in the studied population suggests the importance of information campaigns dedicated to the general population and the relevance of scientific activities with respect to increasing knowledge and awareness of AD among physicians.

Notably, serological markers currently used by physicians to support the diagnosis of AD are limited to total and/or allergen-specific serum IgE levels and peripheral eosinophil counts, which are characterized by low sensibility and specificity [14]. The recent discovery of a new subset of T-cell cytokines and chemokines has resulted in the introduction of multiple potential biomarkers, including serum levels of CD30; macrophage-derived chemoattractant (MDC); interleukins (IL)-12, -16, -18, and -31; and thymus and activation-regulated chemokine (TARC) [14,15,16]. Although none of these novel biomarkers have proven reliable for the diagnosis of AD in clinical practice to date, we consider further research necessary in this field with the aim of simplifying the diagnosis and management of the disease [14].

In the AD subgroup, the mean duration of disease was of 15.3 years, with a mean EASI score of 11.2, IGA score of 1.9, itch-NRS of 6.9 and sleep-NRS of 5.0. Interestingly, AD patients showed the highest values of all patient-reported outcomes in comparison with non-AD patients. Statistically higher severity in sleep disturbances was observed in the AD group vs. 3.9 in non-AD group (*p* < 0.0001). Notably, sleep disturbances are a well-known manifestation associated with AD, with a prevalence ranging between 33% and 81.7% in adults, not exclusively related to itch but also to immunological and neuroendocrine imbalance [17]. A few studies suggested a correlation between poor sleep quality and AD, regardless of disease status, suggesting that repeated flares of AD over time can lead to behavior-related sleep disorders persisting despite disease remission [18]. Thus, high prevalence of sleep disturbances detected in our AD patients could be also explained by the significantly longer duration of disease reported in the AD subgroup (15.6 years) compared to non-AD individuals (6.9 years) (*p* < 0.0001).

Furthermore, chronic sleep disorders have been identified as one of the most important risk factors for the development of several non-atopic comorbidities in AD, including mental health disorders (e.g., anxiety, depression) and cardiovascular diseases (e.g., coronary artery disease and hypertension) [1,2,19]. All these comorbidities may in turn adversely affect sleep quality and increase the disease burden of AD patients [19].

A recent survey including Irish adult patients affected by AD revealed a negative influence of disease on social and relational life, with 70% of patients reporting social anxiety, 65% avoiding sport and physical activities, 52% avoiding social activities, and 52% avoiding sexual intimacy [20]. Similarly, we detected significantly higher social embarrassment and interference with sport activities and job tasks in the AD subgroup compared to the non-AD subgroup, confirming AD as a severely debilitating cutaneous disease with multiple effects on patients’ overall quality of life. The effect of AD on adults and children can currently be determined by different quality of life questionnaires, the most used of which are the Dermatology Quality of Life (DLQI), Children’s Dermatology Quality of Life, and Infants Dermatology Quality of Life questionnaires; it is important to consider that all these tools are not specific for AD [21,22,23].

Considering the extensive and multimodal burden of AD, the development and evaluation of new specific questionnaires to evaluate the multiple domains influenced by AD would be very useful.

Almost one-third of patients did not use any compound, either topical or systemic, whereas only 7.0% of patients were treated with a systemic therapeutic agent. Notably, the economic burden of topical therapies, which are not covered by the national health care system, might negatively impact treatment access. In this study, we found a positive correlation between the monthly expense for topicals and both patient- and physician-assessed disease severity, suggesting that poorly controlled AD requires an increased use of topical agents. This increased use of topical agents could be due to undertreatment, which does not include systemic agents, which are only prescribed in a small percentage of patients (7.0%), notwithstanding the consistent number of subjects suffering from moderate-to-severe AD (EASI ≥ 16 and IGA ≥ 3 reported in 21.5% and 25.3% of patients, respectively).

Importantly, only 2.8% of patients reported complete satisfaction with treatments received to date, showing profound unmet therapeutic needs among adult patients affected by AD [24]. Notwithstanding the recent introduction of novel targeted therapies approved for AD, which can be prescribed by tertiary healthcare centers only, more than 80% of patients reported lack of awareness about the existence of these therapeutic opportunities [25,26,27,28]. This indicates the necessity of creating a proactive network connecting territorial dermatologists to secondary and tertiary centers with the aim of enhancing the therapeutic management of AD patients.

In conclusion, this study underlines the utility of organizing information campaigns on AD to enhance awareness regarding disease features and management and to facilitate early diagnosis with a subsequent reduction in the burden of disease.

## Figures and Tables

**Table 1 jcm-11-05204-t001:** Characteristics of general population, and separately for atopic dermatitis (AD) and non-AD.

	General Population	AD Population	Non-AD Population	*p*-Value *
Patients N tot	641	284	357	
Males n (%)	246 (38.4)	96 (33.8)	150 (42.0)	0.033
Age (mean ± SD)	46.2 ± 19.4	36.9 ± 16.4	53.7 ± 17.9	<0.0001
BMI (mean ± SD)	24.3 ± 4.4	23.5 ± 4.4	25.1 ± 0.23	<0.0001
Previous diagnosis of AD n (%)	362 (56.5)	219 (77.1)	126 (35.3)	<0.0001
Disease duration (mean ± SD)	10.7 ± 11.8	15.3 ± 12.6	7.0 ± 9.7	<0.0001
Family history of AD n (%)	115 (17.9)	66 (23.2)	47 (13.1)	<0.0001
Family history of other atopic comorbidities n (%)	183 (28.5)	109 (38.4)	72 (20.2)	<0.0001
**Job Title**				
Intellectual jobs (white collars) n (%)	260 (40.6)	124 (43.7)	136 (38.1)	0.176
Manual jobs (blue collars) n (%)	154 (24.0)	78 (27.5)	76 (21.3)	0.069
**Physicians Previously Consulted**				
Dermatologist n (%)	542 (84.5)	264 (93.0)	278 (77.9)	<0.0001
Allergologist n (%)	219 (34.2)	139 (48.9)	80 (22.4)	<0.0001
General practitioner n (%)	39 (6.1)	6 (2.1)	33 (9.2)	<0.0001
**Patient’s Related Outcomes**				
Itch intensity (mean, SD)	6.5 ± 2.5	6.9 ± 2.4	6.2 ± 2.6	0.0003
Interference with sleep (mean ± SD)	4.4 ± 3.4	5.0 ± 3.4	3.9 ± 3.4	<0.0001
Interference with work (mean ± SD)	3.7 ± 3.4	4.3 ± 3.5	3.4 ± 3.4	0.002
Interference with sport (mean ± SD)	3.4 ± 3.4	4.1 ± 3.4	2.8 ± 3.2	<0.0001
Disease induced embarassment (mean ± SD)	5.7 ± 3.2	6.4 ± 2.9	5.1 ± 3.3	<0.0001
**Treatments Routinely Used**				
Topical compounds n (%)	259 (40.4)	126 (44.3)	133 (37.2)	<0.0001
Moisturizers n (%)	294 (45.8)	160 (56.3)	134 (37.5)	<0.0001
Systemic therapies n (%)	41 (6.4)	20 (7.0)	21 (5.9)	0.07
No treatments n (%)	225 (35.1)	90 (31.7)	135 (37.9)	<0.0001

Legend. Categorical data expressed as n (%); continuous data expressed as mean ± standard deviation * *p* value refers to the comparison between AD and non-AD population.

**Table 2 jcm-11-05204-t002:** Predictors of average monthly expense > 20 euros for topical therapies, systemic therapies, and visits.

	Monthly Expense for Topical Therapies > 20 Euros OR (95%CI)	Monthly Expense for Systemic Therapies > 20 Euros OR (95%CI)	Monthly Expense for Visits > 20 Euros OR (95%CI)
EASI *	**1.03 (1.01–1.06)**	0.99 (0.96–1.02)	1.01 (0.99–1.03)
IGA scoring *	**1.56 (1.18–2.05)**	1.26 (0.89–1.78)	0.96 (0.72–1.28)
Pruritus (0–10) **	**1.12 (1.01–1.25)**	1.11 (0.94–1.31)	1.02 (0.90–1.15)
Embarassment (0–10) **	**1.19 (1.09–1.31)**	**1.21 (1.04–1.42)**	**1.14 (1.02–1.27)**
Interference with work (0–10) **	**1.15 (1.06–1.26)**	1.10 (0.99–1.24)	1.07 (0.98–1.17)
Interference with sport (0–10) **	**1.19 (1.07–1.32)**	1.09 (0.95–1.24)	1.09 (0.98–1.21)
Interference with sleep (0–10) **	**1.13 (1.04–1.23)**	1.04 (0.93–1.18)	1.06 (0.97–1.17)
Atopic comorbidities **	**1.35 (0.71–2.57)**	1.76 (0.65–4.78)	1.42 (0.69–2.95)
Job (ref: white collar) **			
Blue collar	0.98 (0.49–1.97)	1.88 (0.75–4.72)	1.49 (0.73–3.08)
Retired/unemployed	1.31 (0.47–3.6)	2.15 (0.56–8.28)	0.85 (0.26–2.77)

Legend. * Model adjusted for: age, gender ** Model adjusted for age, gender, EASI. Statistically significant results are highlighted in bold.

## Data Availability

The data relating to this study are available in an excel file.

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
