# Peer review of "National Information Campaign Revealed Disease Characteristic and Burden in Adult Patients Suffering from Atopic Dermatitis"

_jcm, 2022, doi:10.3390/jcm11175204_

Round 1

Reviewer 1 Report

Current study by Gori et al underlines the utility of organizing information campaigns on atopic dermatitis to enhance awareness regarding disease features and management, which facilitate early diagnosis with subsequent reduction of the burden of disease. It is a well written article. I have few comments:

 A separate section should be mentioned for biomarkers that have been described before for  general clinical use for diagnosis of atopic dermatitis e.g. serum levels of CD30, IgE levels, Macrophage-Derived Chemoattractant (MDC), interleukins (IL)-12

 In clinical trials, the most commonly used AD-specific, dermatology-specific, and generic scale for children’s should be mentioned

 Future perspectives for further development and evaluation of practical clinical quality of life scales for AD should be discussed.

 Association of AD with other disease or disorders e.g. cancer, obesity, and its impact on person’s sleep or behavior should be mentioned .

Reviewer 2 Report

The Manuscript jcm-1846954-peer-review-v1

Manuscript jcm-1846954-peer-review-v1 describes the importance of national information campaign to reveal disease characteristics and burden in adult patients suffering from atopic dermatitis.

The manuscript has important findings that could be of interest to the readers of Journal of Clinical Medicine. However, the following comments need to be addressed before publication.    

Major concerns:

1.       Title: the study was designed to elucidate the characteristics AD in adult patients. It is important to amend the title to include “adult patients”.

2.       Abstract, lines 64-66: the sentence “Almost one third of patients ……..  were treated with a systemic treatment” is confusing. It might be clearer to describe the percentage of patients who were treated by topical, systematic, and both topical and systematic, as well as the untreated patients. 

3.       Abstract, lines 68-70: the sentence “This study shows a profound unmet need in AD,” is vague. The authors need to be more specific and describe the main findings followed by a conclusion.

4.       Keywords, line 72: the word “open-day” seems unrelated to the topic of the manuscript. It is recommended to avoid.

5.       Materials and methods: it is unclear if the authors have obtained ethical approval to conduct the study.

6.       Materials and methods: the inclusion and the exclusion criteria were not described.

7.       Materials and methods, line 126: Although it was described in the results section, it is recommended to describe professions that were highlighted by white and blue collars in the materials and method section as it first appears in the manuscript.

Minor concerns:

1.       Abstract, line 61: the following sentence “resulted affected” should read “affected”.

2.       Abstract, lines 63-64: the abbreviated phrases must be mentioned in full as they first appear in the abstract.

3.       Introduction, line 99: “of AD it might be” should read “of AD, it might be”.

4.       Results, line 159: “manifestations resulted of” should read “manifestations was”.

5.       Results, line 190: “37.5% 134/357” should read “37.5% (134/357)”.

Author Response

Dear Reviewer,

thank you for your suggestions.

Below is the list of answers to your questions point by point.

Question 1

Title: the study was designed to elucidate the characteristics AD in adult patients. It is important to amend the title to include “adult patients”.

Answer 1

I included the term "adult patients" in the title

Question 2

Abstract, lines 64-66: the sentence “Almost one third of patients ……..  were treated with a systemic treatment” is confusing. It might be clearer to describe the percentage of patients who were treated by topical, systematic, and both topical and systematic, as well as the untreated patients. 

Answer 2

I described in the abstract the percentage of patients for each type of treatments

Question 3

Abstract, lines 68-70: the sentence “This study shows a profound unmet need in AD,” is vague. The authors need to be more specific and describe the main findings followed by a conclusion.

Answer 3

I described the main findings of the study at the end of the abstract 

Question 4

Keywords, line 72: the word “open-day” seems unrelated to the topic of the manuscript. It is recommended to avoid.

Answer 4

I replaced the term Open-day with information campaign

Question 5

Materials and methods: it is unclear if the authors have obtained ethical approval to conduct the study.

Answer 5

As a screening program, it was not necessary to apply for ethics committee approval, because patients didn’t furnish any sensitive data to the centers.

I added this phrase in the text (127-129)

Question 6

Materials and methods: the inclusion and the exclusion criteria were not described.

Answer 6

I added inclusion criteria in Materials and methods (line 130-131)

Question 7

Materials and methods, line 126: Although it was described in the results section, it is recommended to describe professions that were highlighted by white and blue collars in the materials and method section as it first appears in the manuscript.

Answer 7

I described in materials and methods the professions that were highlighted by white and blue collars (line 137-140)

I followed all suggestions contained in minor concerns.
